# Species Identification and Fungicide Sensitivity of *Fusarium* spp. Causing Peanut Root Rot in Henan, China

**DOI:** 10.3390/jof11060433

**Published:** 2025-06-06

**Authors:** Min Li, Liting Chen, Qinqin Wang, Leiming He, Yun Duan, Xuewei Mao, Lin Zhou

**Affiliations:** 1College of Plant Protection, Henan Agricultural University, No. 218, Ping’an Avenue, Zhengzhou 450046, China; 18838934006@163.com (M.L.); 19836939001@163.com (L.C.); wangqqhau@163.com (Q.W.); leimingh@henau.edu.cn (L.H.); 2Henan Key Laboratory of Creation and Application of New Pesticide, Henan Agricultural University, Zhengzhou 450046, China; 3Henan Research Center of Green Pesticide Engineering and Technology, Henan Agricultural University, Zhengzhou 450046, China; 4Institute of Plant Protection Research, Henan Academy of Agricultural Sciences, Zhengzhou 450002, China; duanyunhao@163.com

**Keywords:** peanut root rot, pathogen identification, *Fusarium* species distribution, pathogenicity, fungicide sensitivity

## Abstract

Peanut Root Rot (PRR) is a devastating disease that significantly limits peanut production worldwide. Although PRR has been frequently reported in Henan Province of China, the predominant *Fusarium* species and their sensitivity to different fungicides remain unclear. Between 2021 and 2023, we surveyed 81 peanut fields across 17 cities in Henan Province, China, to assess PRR prevalence and *Fusarium* species distribution. A total of 1131 *Fusarium* isolates were identified based on the morphological characters and phylogenetic analyses and classified into 11 recognized *Fusarium* species: *F. solani* (56.06%), *F. oxysporum* (20.87%), *F. neocosmosporiellum* (13.62%), *F. proliferatum* (4.69%), *F. acuminatum* (1.33%), *F. commune* (1.15%), *F. graminearum* (1.06%), *F. pseudograminearum* (0.35%), *F. ipomoeae* (0.35%), *F. lacertarum* (0.26%), and *F. armeniacum* (0.26%). Pathogenicity assessments showed that all 11 *Fusarium* species were capable of causing PRR, with *F. solani* exhibiting the highest isolation frequency and widespread distribution in all areas. Furthermore, the four *Fusarium* species (*F. solani*, *F. oxysporum*, *F. neocosmosporiellum*, and *F. proliferatum*) were highly sensitive to the six fungicides, including prochloraz (EC_50_ values of 0.02 ± 0.00~0.06 ± 0.01 mg/L), pydiflumetofen (EC_50_ values of 0.31 ± 0.07~0.67 ± 0.06 mg/L), tetramycin (EC_50_ values of 0.11 ± 0.02~0.58 ± 0.08 mg/L), tebuconazole (EC_50_ values of 0.26 ± 0.07~0.65 ± 0.10 mg/L), prothioconazole (EC_50_ values of 1.14 ± 0.16~3.15 ± 0.81 mg/L), and difenoconazole (EC_50_ values of 0.62 ± 0.12~3.58 ± 0.76 mg/L). This comprehensive study is the first systematic documentation on the prevalence, virulence, and fungicide sensitivity of PRR pathogens in Henan Province. The findings of the current study will provide a theoretical basis for the effective management of peanut root rot in Henan, China.

## 1. Introduction

Peanut (*Arachis hypogaea* L.) is an important oil and economic crop widely cultivated in tropical and subtropical agro-climatic regions [1]. China is the largest producer of peanuts in the world, with a peanut planting area of 4,850,656 ha in 2023, producing 19.27 MT peanuts, accounting for approximately 35.51% of the global peanut yield (54.27 MT) [2]. China’s peanut growing areas are mainly distributed in the Henan Province, which contributes significantly to the national peanut yield and planting area at 33.22% and 27.24%, respectively (http://www.stats.gov.cn, accessed on 10 February 2025). In recent years, due to the adjustment of the cropping system and changes in climate conditions, peanut root rot (PRR) has been frequently and widely reported in Henan Province, posing a serious threat to the peanut yield and quality [3].

PRR is a devastating disease affecting peanut crops, which has been found in peanut fields in Australia [4], Vietnam [5], Pakistan [6], Argentina [7], Egypt [8], India [9], Ethiopia [10] and China [11,12,13,14]. The early symptoms of the disease are brown spots on the root and wilting on the lower leaves [12,13,15]. As the disease progresses, infected peanut plants exhibit stunted, yellowing leaves, diminished plant vigor or sudden wilting, and the presence of brown/black discoloration and rot on the infected roots [4,11,12,13]. Eventually, the affected plants collapse and die [11,12,13,15]. The disease usually causes from 20% to 40% of disease incidence in peanut fields [4,13,14]. In serious cases, the incidence of this disease was found in up to 75% in peanut fields [10]. Especially during seasons with continuous drought stress, this disease could cause a 95% disease incidence in some peanut fields [16].

PRR is caused by multiple pathogenic fungal species [6,7,8,9,11,12,17,18]. Among them, various *Fusarium* species have been implicated in PRR worldwide, with *F. solani* and *F. oxysporum* being the most frequently reported pathogens across multiple regions, including Argentina [7], Egypt [8,19], Ethiopia [10], and so on. However, the *Fusarium* species of PRR are remarkably varied with different geographical regions. In Egypt, *F. solani*, *F. moniliforme*, *F. equesti*, *F. semitectum*, and *F. oxysporum* have been reported to cause root rot in peanut fields [8,19]. In Argentina, *F. oxysporum* and *F. solani* have been implicated as responsible for PRR [7]. In India, *F. neocosmosporiellum* and *F. solani* are isolated from diseased peanut plants, which caused PRR [20]. In Australia and Vietnam, PRR disease is caused by *F. neocosmosporiellum* [4,5]. In eastern Ethiopia, *F. oxysporum* and *F. solani* are isolated from peanut roots, and identified as PRR pathogens [10]. In China, *F. equiseti*, *F. incarnatum*, *F. oxysporum*, *F. proliferatum*, *F. solani*, *F. fujikuroi*, and *F. acuminatum* have been identified as PRR pathogens [13,14,21,22]. However, it remains uncertain which specific *Fusarium* species are most prevalent and play a significant role in causing PRR in Henan Province.

A thorough comprehension of pathogens involved in PRR is essential for effective disease management. Previous studies have identified *Fusarium* species as causative agents of PRR in various provinces of China, including Jiangxi [21], Shandong [14], and Henan [13,22]. However, comprehensive surveys of PRR in these provinces, especially in Henan Province, are lacking. In present, only two species (*F. fujikuroi* and *F. proliferatum*) have been reported in Henan Province [13]. In a recent investigation, 41 fungal isolates were collected from the diseased peanut roots in six cities within Henan Province (Anyang, Jiaozuo, Pingdingshan, Xinxiang, Xuchang, and Zhoukou). These isolates were identified to *F. equiseti*, *F. incarnatum*, *F. oxysporum*, *F. proliferatum*, and *F. solani*, respectively, all of which were confirmed to be the pathogens of PRR [22]. However, due to the limited sampling areas and a small number of isolates, these data may not accurately reflect the prevalence and distribution of pathogens associated with PRR in Henan Province. Therefore, a comprehensive understanding of biology, population dynamics, and epidemiological factors contributing to disease exacerbation, is necessary for effective disease control strategies.

At present, various management measures have been reported to be effective approaches for control of PRR, such as chemical control [23,24], biological control [15,16,17], and agricultural practices like crop rotation or intercropping [21,25]. However, the application of fungicides remains the more effective approach for disease control. Although different fungicides have been widely used to control PRR, there is limited information on the sensitivity of PRR-causing pathogens to fungicides, particularly in China. To effectively manage PRR, it is crucial to understand the sensitivity of local pathogens to various fungicides utilized to manage this disease.

Therefore, the main objectives of this research were to (i) identify the *Fusarium* species causing root rot on peanut in the major peanut-cultivation regions of Henan; (ii) determine the virulence of each *Fusarium* species obtained from peanut roots; and (ⅲ) assess the inhibitory effect of registered and other alternative potential fungicides against mycelial growth of the four *Fusarium* species (*F. solani*, *F. oxysporum*, *F. neocosmosporiellum*, and *F. proliferatum*), the most frequent pathogens of PRR in the current study.

## 2. Materials and Methods

### 2.1. Sample Collection

During 2021 to 2023, peanut samples with typical root rot symptoms were collected from the seedling stage to flowering stage in Henan Province, China. A total of 81 peanut fields in 17 cities (including Puyang, Anyang, Hebi, Xinxiang, Jiaozuo, Jiyuan, Luoyang, Zhengzhou, Kaifeng, Shangqiu, Zhoukou, Xuchang, Luohe, Pingdingshan, Nanyang, Zhumadian and Xinyang) were investigated, representing the main peanut-producing regions in Henan Province (Figure 1). The peanut fields were arbitrarily selected, with a minimum distance of 5 km between each other. For each peanut field, five sampling points were made in a zig zag pattern [26], with each point being 20 m apart. At each of the five points, approximately 20 peanut plants were assessed, and on return to the laboratory 3–4 plants with typical symptoms of root rot were randomly selected from each transect point, for a total of 15–20 plants per field. All samples obtained were stored at 4 °C before fungal isolation.

### 2.2. Isolation of Fungi and Morphological Observation

The infected peanut roots were washed with running water, dried on filter paper, and then 5 mm length pieces were taken from diseased to healthy tissue. The pieces were surface-disinfected for 1 min with 3% NaClO solution, rinsed three times in sterile distilled water, and dried on sterilized filter paper. Five pieces of each root were placed on potato dextrose agar (PDA) plates containing ampicillin (100 mg/L). These plates were incubated at 25 °C in the dark for 2 days. When fungal colonies appeared, they were chopped using sterile toothpicks and transferred to the new PDA plates. Each isolate obtained were further purified by using the single spore isolation method [27]. All pure fungal isolates obtained in the present study were stored in PDA tubes at 4 °C.

To confirm whether these fungal isolates were *Fusarium* species, they were preliminarily identified based on morphological characteristics as described in the literature [28,29]. Briefly, all pure fungal isolates were cultured on PDA plates and colonies morphology were observed after 7 days of incubation in the dark at 25 °C. Furthermore, all pure fungal isolates were cultured in carboxymethyl cellulose medium (CMC) to induce sporulation in darkness at 25 °C, 175 rpm and micromorphological characteristics (microconidia, macroconidia, and conidiophores) were observed using a Zeiss Imager M2 compound microscope with differential interference contrast (Carl Zeiss AG, Oberkochen, Germany) with the Zeiss software ZEN 3.4 after 3 days.

### 2.3. DNA Extraction, PCR Amplification and Sequencing

Genomic DNA of all isolates were extracted using a modified CTAB method as described by Özer et al. [30]. Five gene fragments, including ITS, *TEF-1α*, *RPB2*, *TUB2*, and *CAM* of 49 representative isolates (Appendix A), were amplified with the primer pairs ITS1/ITS4 [31], EF1/EF2 [32], 5f2/7cr [33,34], T1/T2 [35], and CL1/CL2A [36], respectively. For the other isolates, only the *TEF-1α* gene region was sequenced. The primer pairs used in the study are shown in Table 1. DNA amplification was conducted in a total volume of 25 μL reaction solution that contained 22 μL 1.1 × T3 Super PCR Mix (Tsingke Biotech Co. Ltd., Beijing, China), 1 μL of each primer (10 mM), and 1 μL template DNA (50 ng/μL). The PCR cycling conditions were initiated with 98 °C for 2 min, followed by 35 cycles of denaturation at 98 °C for 10 s, annealing at a suitable temperature for 10 s for different loci: 56 °C for ITS, 55 °C for *TEF-1α*, 58 °C for *RPB2*, 54 °C for *TUB2*, and 55 °C for *CAM*, and extension at 72 °C for 10 s, and a final elongation step at 72 °C for 2 min. The PCR products were separated by agarose gel electrophoresis (1%, wt/vol) and sequenced by Tsingke Biotech Co. Ltd., Beijing. The obtained sequences were examined and aligned using BioEdit version 7.0.5.2 [37]. For the *TEF-1α* gene region, a sequence match of more than 99% similarity was regarded as the indicative of a fungal species in the present study [38].

### 2.4. Phylogenetic Analysis

Based on morphology and *TEF-1α* sequence data, 49 isolates were selected to perform the phylogenetic analysis, including 5 *F. solani* isolates, 5 *F. oxysporum* isolates, 5 *F. neocosmosporiellum* isolates, 5 *F. proliferatum* isolates, 5 *F. acuminatum* isolates, 5 *F. commune* isolates, 5 *F. graminearum* isolates, 4 *F. pseudograminearum* isolates, 4 *F. ipomoeae* isolates, 3 *F. lacertarum* isolates and 3 *F. armeniacum* isolates (Appendix A). Multiple sequence alignments for each of five gene sequence fragments (ITS, *TEF-1α*, *RPB2*, *TUB2*, and *CAM*), including sequences generated from the present study and sequences from related taxa (Appendix A), were initially implemented using the MAFFT version 7.110 [39] with default parameters setting before being manually adjusted using BioEdit version 7.0.5.2 [37]. The phylogenetic analysis was performed by two independent algorithms, including Maximum likelihood (ML) and Bayesian inference (BI). The ML analysis was executed in the IQ-TREE version 1.6.8 [40] utilizing the GTR model with 1000 ultrafast bootstrap (BS) replications. Bayesian inference (BI) was utilized to construct phylogenetic relationship via MrBayes version 3.2.6 [41]. The best nucleotide substitution model of each partition was created by MrModeltest version 2.3 [42]. Markov Chain Monte Carlo (MCMC) analysis with four chains was conducted twice from a random tree topology for 3,000,000 generations. Trees were sampled every 1000 generations, which lasted until the average standard deviation of split frequencies was <0.01. After discarding the first 25% of saved trees (burn-in), the remaining trees were used to calculate the 50% majority rule consensus trees and posterior probability (PP) values. Clades with PP ≥ 0.95 and bootstrap values (BS) ≥ 70% were regarded as well supported [43]. Phylogenetic trees were estimated and adjusted using FIGTREE version 1.4.2 (http://tree.bio.ed.ac.uk/software/figtree/, accessed on 25 May 2025). *Fusarium ventricosum* CBS 748.79 served as the outgroup taxon in the phylogenetic analysis of *Fusarium* spp. The GenBank numbers of all isolates chosen for the phylogenetic analyses are shown in Appendix A.

### 2.5. Prevalence

The prevalence of fungal species in the main peanut-producing regions of Henan province was estimated as previously described [44]. The Isolation Frequency (FI) of individual species was calculated by the formula, FI% = (NS/NI) × 100, where NS was the number of isolates per fungal species, and NI was the total number of isolates from all fungal species obtained.

### 2.6. Pathogenicity Tests

Forty-nine isolates from 11 *Fusarium* species (Appendix A) were selected to evaluate their pathogenicity on peanut seedlings by a modified inoculum layer method [45]. Briefly, all isolates selected were grown on PDA at 25 °C for 14 d, and the spores were suspended in 5 mL of sterile deionized water by scraping the colony with a sterile inoculating loop, filtered, then adjusted to the concentrations of 10^6^ conidia per mL. The triangular flasks (250 mL) containing 50 g autoclaved millet seeds were inoculated with 2 mL spore suspension (10^6^ conidia per mL), which were incubated at 25 °C for 7 days and were shaken daily for 30 s to ensure uniform colonization. Seeds of peanut (cv. Yuhua 22, a widely cultivated variety in Henan Province) were soaked for 1 min in 75% alcohol, rinsed in sterile water, and then germinated on sterile paper towels saturated with sterile water for 2 days at 25 °C. For planting, a 2 g layer of millet seed inoculums was placed in a plastic pot (7 cm × 7 cm × 8 cm) containing 15 g of sterilized premium grade coarse dry vermiculite and then covered with 8 g of vermiculite. Subsequently, two germinated peanut seeds with healthy and consistent growth were selected and sown on the top of the vermiculite, and 8 g layer of vermiculite was added to the top of the seeds. Finally, the sterilized water (80 mL) was added slowly to the pots to saturate the vermiculite. The sterile millet seed without mycelia was regarded as the negative control. Each four pots were used as a replicate unit, with three replications of each isolate tested. The experiments were repeated in triplicate. All pots were provided with natural daylight and water regularly, in a greenhouse with a temperature of 25–30 °C. At 3 weeks after of planting, the peanut seedlings were carefully taken out from the vermiculite and roots washed under tap water. The severity of peanut root rot symptoms was recorded using a 0 to 4 rating scale, as in previous studies, with a slight modification [16,46], where 0 = no symptoms, 1 = mild symptoms (discoloration but no visible lesions), 2 = obvious lesions (severe discoloration with lateral root reduction), 3 = severe lesions on the taproot and lateral root and diminished plant vigor, and 4 = hypocotyl rotten, plant dead. Disease severity index (DI) was calculated using the following equations [46]: DI = [Σ (number of diseased peanut plant in each scale × disease scale)/(the total number of observed peanut plants × highest scale)] × 100. The pathogenic degree of the different fungal species was assessed according to Bertoldo et al. [47]’s descriptions, namely non virulence (N), DI = 0 to 10: low virulence (L), DI = 11 to 30; moderate virulence (M). DI = 31 to 60; and high virulence (H), DI = 61 to 100.

For each isolate, three inoculated roots with diseased symptoms were selected to confirm Koch’s postulates. The pathogens were re-isolated from diseased root tissues as described above. To determine whether the recovered isolates were the same species as those used for inoculations, morphological observations and *TEF-1α* sequencing of the recovered isolates were conducted and compared against the inoculated isolates.

### 2.7. Fungicide Sensitivity Assays

Tebuconazole (97.30% a.i., Guangxi Tianyuan Biochemical Co., Ltd., Nanning, China), difenoconazole (98.40% a.i., Qingdao Taisheng Biological Technology Co., Ltd., Qingdao, China), prothioconazole (96.00% a.i., Guangxi Tianyuan Biochemical Co., Ltd.), prochloraz (96.50% a.i., Qingdao Taisheng Biological Technology Co., Ltd.), tetramycin standard (15.00% a.i., Liaoning Wkioc Bioengineering Co., Ltd., Chaoyang, China), pyraclostrobin (98.60% a.i., Qingdao Taisheng Biological Technology Co., Ltd.), and pydiflumetofen (99.66% a.i., Beijing Qincheng Yixin Technology Development Co., Ltd., Beijing, China) were dissolved in dimethyl sulfoxide (DMSO) to prepare stock solutions of 10,000 mg/L.

The antifungal activity of seven fungicide against four fungal species (*F solani*, *F. oxysporum*, *F. neocosmosporiellum* and *F. proliferatum*) was determined using a mycelium growth rate method in vitro. Mycelial plugs (5 mm in diameter) cut from the edge of 7-day-old colonies were transferred to PDA plates amended with different concentrations for each of the fungicides (Appendix A). PDA plates containing the same volume of DMSO were used as controls. For pyraclostrobin, salicylhydroxamic acid (SHAM) was added to the fungicide-amended/fungicide-free PDA plates at a final concentration of 100 μg/mL to block the alternative oxidase pathway [48]. The above PDA plates were incubated at 25 °C in the dark. When colonies in the control treatment covered the entire plate area, colony diameter under each treatment was measured by the cross method, and the mycelial growth inhibition rate was calculated according to the formula: inhibition rate = [(the diameter of control-5 mm) − (the diameter of treatment-5 mm)]/(the diameter of control-5 mm) × 100. The EC_50_ (the effective concentration for 50% inhibition) values of the fungicides was calculated by linear regression analysis of the percentage of growth inhibition plotted against the log10 fungicide concentrations. Five representative isolates from each fungal species were designated for the assay (Appendix A). Three replicate plates were used for each isolate/fungicide/combination, and all the experiments were performed in triplicate.

### 2.8. Statistical and Analysis

All data were analyzed using SPSS 18.0 software (SPSS Inc., Chicago, IL, USA). Data were checked for normality (Shapiro–Wilk test for normality, *p* > 0.05) and equality of variances (Levene’s Test, *p* > 0.05) prior to statistical analysis. Differences among different treatments were determined by analysis of variance (ANOVA), and mean values were compared by Tukey’s HSD test at the 5% significance level. Data are presented here as means ± SDs (standard deviation).

## 3. Results

### 3.1. Collection of Fungal Isolates

In the surveyed peanut field, PRR showed the symptoms including chlorotic leaves, stunting and wilting on the above-ground parts of the diseased peanut plants (Figure 2a–c), and light brown to black lesions on the infected root tissues (Figure 2d–g). From these peanut fields, 1555 PRR samples were collected and 1131 *Fusarium* isolates were obtained. Other fungal species were also isolated, including *Rhizoctonia solani* (n = 19 isolates), *Aspergillus niger* (n = 14 isolates), *Rhizopus arrhizus* (n = 3 isolates), and *Lasiodiplodia theobromae* (n = 2 isolates). Nevertheless, due to the predominance and diversity of *Fusarium* spp. that are associated with root rot of peanut, other fungal species were not included in this work.

### 3.2. Species Identification and Phylogenetic Analysis

Based on the colony morphology, these 1131 *Fusarium* isolates were preliminarily classified into 11 *Fusarium* species (Figure 3). For further molecular verification, the phylogenetic tree based on the combined sequences of ITS–*TEF-1α*–*RPB2*–*TUB2*–*CAM* gene partition was constructed. The concatenated ITS, *TEF-1α*, *RPB2*, *TUB2*, and *CAM* sequences dataset involved 83 isolates, including 49 *Fusarium* isolates from this study, 33 reference *Fusarium* isolates and the outgroup *Fusarium ventricosum* (CBS 748.79) (Appendix A). The concatenated dataset contained 2980 characters including gaps (467 for ITS, 670 for *TEF-1α*, 722 for *RPB2*, 442 for *TUB2*, and 679 for *CAM*). In BI analyses, the GTR + I + G model of evolution was selected for ITS and *TEF-1a*, the GTR + G model for *RPB2*, the HKY + G model for *TUB2*, and the SYM + I + G model for *CAM*. The phylogenetic trees from BI and ML analyses showed congruent topological structure. The phylogenetic trees revealed that these *Fusarium* isolates also clustered together with 11 species, which were consistent with the morphological result (Figure 4). Thus, based on morphological and phylogenetic trees, these isolates were identified as *F. armeniacum*, *F. pseudograminearum*, *F. graminearum*, *F. acuminatum*, *F. commune*, *F. ipomoeae*, *F. lacertarum*, *F. neocosmosporiellum*, *F. solani*, *F. oxysporum*, and *F. proliferatum* (Figure 3 and Figure 4).

### 3.3. Prevalence of Fungal Species

Prevalence analyses revealed that *F. solani* (634 isolates, 56.06% of the total isolates, obtained from all 17 sampled areas), *F. oxysporum* (236 isolates, 20.87%, isolated from 16 sampled areas except for Zhoukou), and *F. neocosmosporiellum* (154 isolates, 13.62%, isolated from all 17 sampled areas) were the most frequent species associated with PRR in Henan Province, followed by *F. proliferatum* (53 isolates, 4.69%, isolated from Xinyang, Nanyang, Shangqiu, Pingdingshan, Luoyang, Luohe, Puyang, Zhumadian and Hebi), *F. acuminatum* (15 isolates, 1.33%, isolated from Xinyang, Nanyang, Puyang, and Hebi), *F. commune* (13 isolates, 1.15%, isolated from Xinyang, Nanyang, Zhoukou, and Zhumadian), and *F. graminearum* (12 isolates, 1.06%, isolated from Xinyang, Nanyang, Puyang, and Zhumadian) (Figure 5a,b). The remaining four *Fusarium* species, including *F. pseudograminearum* (four isolates, 0.35%), *F. ipomoeae* (four isolates, 0.35%), *F. lacertarum* (three isolates, 0.26%) and *F. armeniacum* (three isolates, 0.26%), were detected in a less than 1% prevalence and only found in one or two areas (Figure 5a,b).

### 3.4. Pathogenicity Tests

The fungus of 11 *Fusarium* species were used to conduct the pathogenicity tests. After 21 days, peanuts inoculated with each of the 11 *Fusarium* species exhibited typical symptoms of PRR, similar to the symptoms initially observed on naturally infected peanut roots, while no symptoms appeared on the negative controls (Figure 6a–l). All *Fusarium* isolates of aforementioned test were successfully re-isolated from the infected peanut roots to satisfy Koch’s postulates. Although these *Fusarium* species can infect peanut, they exhibited the different symptoms on infected roots of peanut (Figure 6b–l). The symptoms of *F. lacertarum* and *F. ipomoeae* showed as light brown to dark brown discoloration on the infected root tissues (Figure 6b,c); *F. armeniacum* and *F. acuminatum* were able to induce brown discoloration with conspicuous black necrotic spots (Figure 6d,e). *F. pseudograminearum*, *F. graminearum* and *F. commune* could cause brown/black lesions with secondary root reduction (Figure 6f–h). *F. solani*, *F. oxysporum*, *F. neocosmosporiellum* and *F. proliferatum* produced brown/black lesions with main and lateral root reduction (Figure 6i–l).

In addition, based on the calculated DI, these pathogenic fungal species showed different levels of virulence (Figure 7). *F. proliferatum*, *F. neocosmosporiellum*, *F. oxysporum*, and *F. solani* were rated as the highly virulent pathogens, but the DI of *F. proliferatum* was significantly greater than those of *F. oxysporum and F. solani* (*p* < 0.05). *F. commune*, *F. graminearum*, *F. pseudograminearum*, *F. acuminatum*, *F. armeniacum* and *F. ipomoeae* were considered to be moderately virulent, whereas *F. commune* exhibited a significantly higher DI compared with *F. acuminatum*, *F. armeniacum* and *F. ipomoeae* (*p* < 0.05). The remaining species *F. lacertarum* was determined to have low virulence.

### 3.5. Fungicide Sensitivity Assays

To identify effective fungicides for controlling PRR, seven fungicides were classified into five classes (imidazoles, antibiotics, demethylation inhibitors [DMIs], quinone outside inhibitors [QoIs], and succinate dehydrogenase inhibitors [SDHIs]) and evaluated in this study. First, four *Fusarium* species, which exhibited high isolation frequency, wide geographical distribution and high virulence, were selected to perform the sensitivity assay. All seven fungicides had different degrees of inhibitory effect on the mycelial growth of the four *Fusarium* species, and the average EC_50_ values ranged from 0.02~27.49 mg/L (Figure 8; Appendix A). Based on the EC_50_ values of the different fungicides, the four *Fusarium* species were sensitive to prochloraz (0.02 ± 0.00~0.06 ± 0.01 mg/L), pydiflumetofen (0.31 ± 0.07~0.67 ± 0.06 mg/L), tetramycin (0.11 ± 0.02~0.58 ± 0.08 mg/L), tebuconazole (0.26 ± 0.07~0.65 ± 0.10 mg/L), prothioconazole (1.14 ± 0.16~3.15 ± 0.81 mg/L) and difenoconazole (0.62 ± 0.12~3.58 ± 0.76 mg/L) (Figure 7). Based on the different *Fusarium* species, the sensitivities of these seven fungicides significantly varied by species (*p* < 0.05) (Figure 8). *F. proliferatum* exhibited a higher sensitivity to prochloraz and difenoconazole compared to *F. solani*, *F. oxysporum* and *F. neocosmosporiellum* (*p* < 0.05), and *F. proliferatum* had less sensitivity to tetramycin and prothioconazole compared with *F. solani* and *F. neocosmosporiellum* except for *F. oxysporum* (*p* < 0.05). The EC_50_ values of *F. neocosmosporiellum* to tebuconazole and pydiflumetofen was separately higher than those of the other three species, respectively (*p* < 0.05), and the EC_50_ values of *F. oxysporum* and *F. proliferatum* to pyraclostrobin were lower than those of the other two species (*p* < 0.05).

## 4. Discussion

Previous studies have shown that the pathogens of PRR are mainly associated with *Fusarium* species [8,10,19,22]. In 2021 and 2023, a total of 1131 *Fusarium* isolates were collected from 17 cities in Henan, an area known for its high incidence of PRR. Based on colony morphology and phylogenetic analyses, the identified *Fusarium* isolates were classified into 11 distinct species, including *F. armeniacum*, *F. pseudograminearum*, *F. graminearum*, *F. acuminatum*, *F. commune*, *F. ipomoeae*, *F. lacertarum*, *F. neocosmosporiellum*, *F. solani*, *F. oxysporum*, and *F. proliferatum* (Figure 3 and Figure 4), and Koch’s postulates confirm that the isolates are pathogenic. Interestingly, the frequency and composition of these *Fusarium* species differed from previous reports [10,20,22], with *F. solani* being the most prevalent species, accounting for 54.23% of PRR pathogens in the region, followed by *F. oxysporum*. Conversely, Debele et al. [10] found that *F. oxysporum* is the most frequent species associated with PRR in eastern Ethiopia, followed by *F. solani*. The reasons for the difference may be due to the climatic conditions, the geographical locations, tillage patterns and peanut variety. Notably, *F. neocosmosporiellum* was identified across major peanut-growing areas in Henan, with a 13.17% isolation frequency, aligning with previous results of its association with diseased peanut roots [20,49]. This study is also the first record of *F. neocosmosporiellum* as a PRR pathogen in Henan. *F. proliferatum* was another common pathogen among these *Fusarium* species, with 53 isolates collected from nine cities, further confirming its presence in the root tissues of peanuts in Henan [13,22]. The remaining seven species, including *F. pseudograminearum*, *F. graminearum*, *F. commune*, *F. ipomoeae*, *F. lacertarum*, *F. armeniacum* and *F. acuminatum*, were detected with a low occurrence in Henan. Notably, *F. acuminatum* had been reported as a PRR pathogen in Shandong, China [14], while the other six *Fusarium* species were first recorded as PRR pathogens globally. Moreover, *F. equiseti* and *F. incarnatum* had also been reported as PRR pathogens in Henan [22]. However, they were not detected during this survey. This may be related to the fact that the investigations of Pan et al. [22] were conducted in the pudding period to the full fruit maturity period, while our survey was carried out during the seedling stage to the flowering stage.

This study showed that different *Fusarium* species displayed different aggressiveness on the peanut roots. Among all tested *Fusarium* species, *F. proliferatum* and *F. neocosmosporiellum* were determined to be highly virulent pathogens, which complemented the previous studies by other researchers [4,5,13,22], who only described that they were pathogenic and responsible for PRR by the Koch’s postulates. Additionally, *F. oxysporum* and *F. solani* were also identified as highly pathogenic to peanut roots. The results corroborate the previous studies [10,19], which found that *F. oxysporum* and *F. solani* could lead to severe root rot of peanut. Furthermore, although *F. commune*, *F. graminearum*, *F. pseudograminearum*, *F. acuminatum*, *F. armeniacum* and *F. ipomoeae* had the lowest frequency of occurrences, these species exhibited moderate virulence on peanut roots. This indicated that these species may be an emerging threat to peanut production in Henan and need be closely monitored in the future. The remaining *Fusarium* species, *F. lacertarum*, exhibited low virulence on the peanut roots. However, it was infrequently isolated in the present study and was considered to be of secondary importance as a pathogen of PRR.

No crop completely immune to the various isolates of *Fusarium* spp. has been reported so far. At present, chemical control is still considered as the most effective and important management strategy for controlling the root rot disease. To date, several fungicides have been registered against PRR in China, including prochloraz, carbendazim, azoxystrobin, pyraclostrobin, difenoconazole, tetramycin, thifluzamide, and so on (http://www.icama.org.cn, accessed on 10 February 2025). However, carbendazim-resistant *F. graminearum* populations have increased in the field in recent years [50]. Succinate dehydrogenase inhibitors (SDHIs) have been applied to manage various plant fungal diseases, but most show poor activity against *Fusarium* spp. [51]. Therefore, to identify suitable fungicides for controlling the most frequent pathogens of PRR in the current research, including *F. oxysporum*, *F. neocosmosporiellum*, *F. proliferatum* and *F. solani*, seven fungicides with different modes of action were evaluated in this study. Our results showed that all species were highly sensitive to prochloraz, tetramycin, and difenoconazole. This result indicated that prochloraz, tetramycin, and difenoconazole can continue to be applied to the management of PRR caused by the four *Fusarium* species. Considering that there are few registered fungicides for controlling PRR, we also determined the sensitivity of the four *Fusarium* species to three unregistered fungicides (tebuconazole, prothioconazole, and pydiflumetofen). The results showed that these three fungicides have great application prospects in the control of *F. oxysporum*, *F. neocosmosporiellum*, *F. proliferatum* and *F. solani*. However, the effects of different fungicides on *Fusarium* spp. under field conditions remain unknown and necessitate further evaluation through pot or field experiments.

## 5. Conclusions

In conclusion, we firstly demonstrated that *F. solani* is the most prevalent species associated with PRR in Henan province, China. In addition, it is the first report that *F. armeniacum*, *F. pseudograminearum*, *F. graminearum*, *F. commune*, *F. ipomoeae*, and *F. lacertarum* occur as pathogens in peanut roots globally. Moreover, it is also the first documentation of *F. neocosmosporiellum* and *F. acuminatum* as a pathogen of PRR in Henan. Fungicide sensitivity assays showed that *F. oxysporum*, *F. neocosmosporiellum*, *F. proliferatum* and *F. solani* exhibited higher sensitivity to prochloraz, tetramycin, tebuconazole, prothioconazole, difenoconazole, and pydiflumetofen compared to pyraclostrobin. Overall, the results of this study provide crucial information for the effective management of PRR in Henan Province. Further work is needed to monitor the development of resistance to these fungicides for successful control of PRR in Henan Province.

## Figures and Tables

**Figure 1 jof-11-00433-f001:**
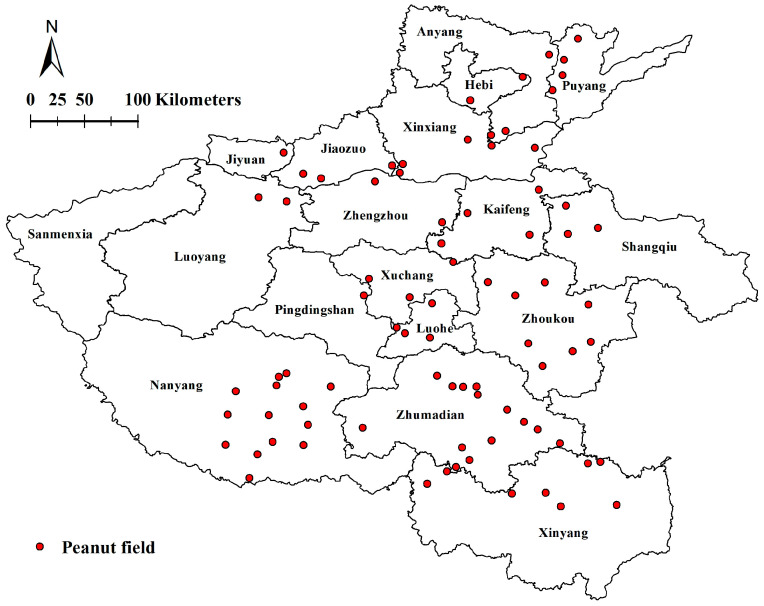
The sample distribution map of peanut fields in the different peanut-producing regions of Henan Province in 2021 to 2023.

**Figure 2 jof-11-00433-f002:**
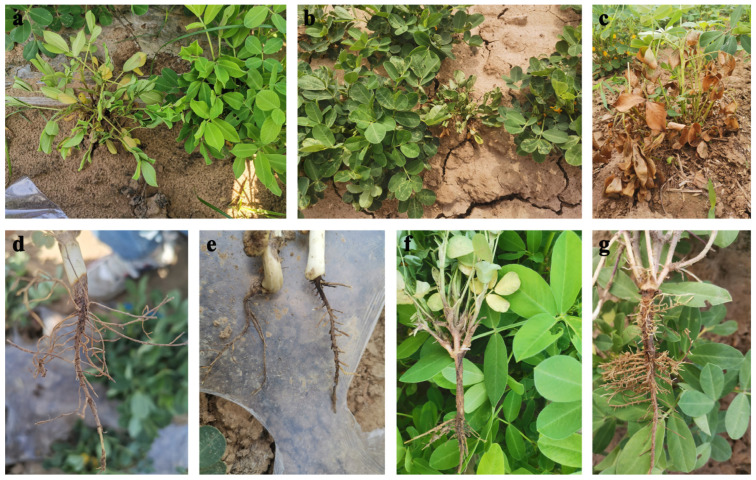
Representative symptoms of peanut root rot in the field. (**a**–**c**) the above-ground parts of diseased peanut plants; (**d**–**g**) the infected root tissues.

**Figure 3 jof-11-00433-f003:**
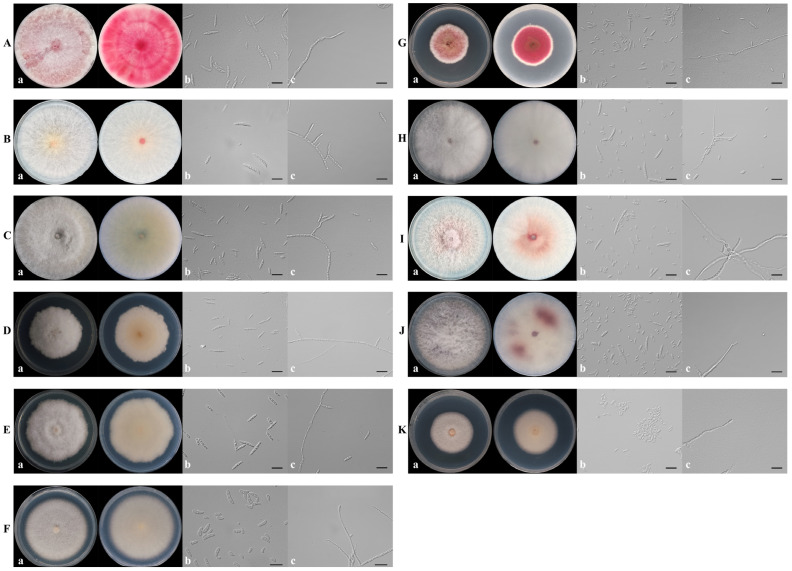
Typical colonies of 11 *Fusarium* species observed after culture on PDA medium at 25 °C for 7 d and conidia and conidiophores in carboxymethyl cellulose (CMC) medium after incubation for 3 d at 25 °C, 175 rpm. (**A**) *F. pseudograminearum*. (**B**) *F. graminearum*. (**C**) *F. armeniacum.* (**D**) *F. ipomoeae.* (**E**) *F. lacertarum.* (**F**) *F. solani.* (**G**) *F. acuminatum.* (**H**) *F. commune.* (**I**) *F. oxysporum.* (**J**) *F. proliferatum.* (**K**) *F. neocosmosporiellum.* The images from a to c were colony morphology, macro- or microconidia, and conidiophores. Scale bar = 20 μm.

**Figure 4 jof-11-00433-f004:**
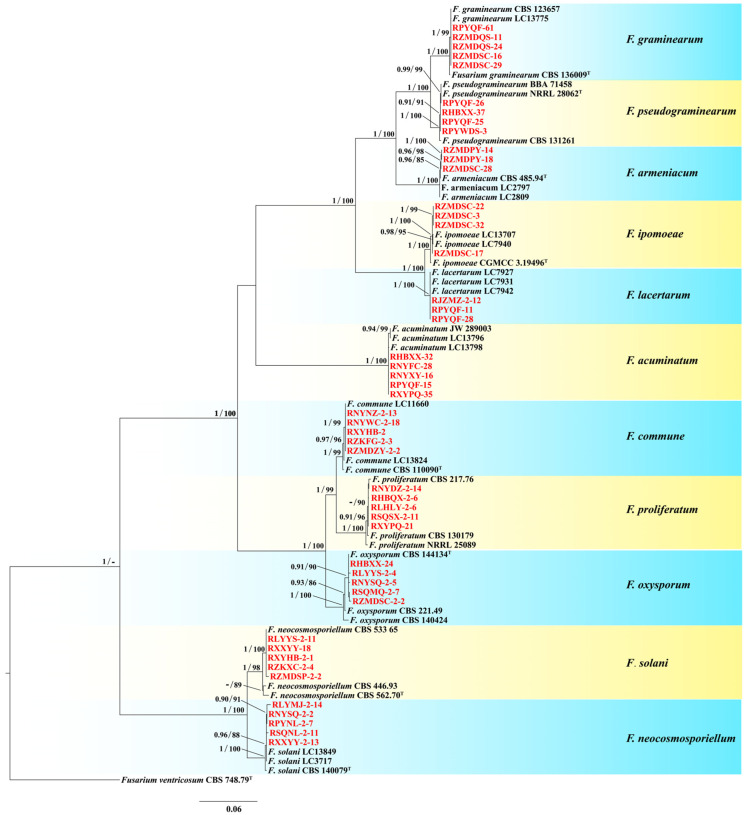
Phylogenetic tree based on Bayesian inference (BI) using MrBayes with the combined sequences of ITS–*TEF-1α*–*RPB2*–*TUB2*–*CAM* gene partition of *Fusarium* isolates. Bayesian posterior probability (PP ≥ 0.90) and IQtree bootstrap support values (BS ≥ 70%) are shown at the nodes (PP/BS). Ex-type isolates were indicated with superscript ‘‘^T^”. Isolates obtained from the present study are indicated in red font. Scale bar indicates the expected number of changes per site.

**Figure 5 jof-11-00433-f005:**
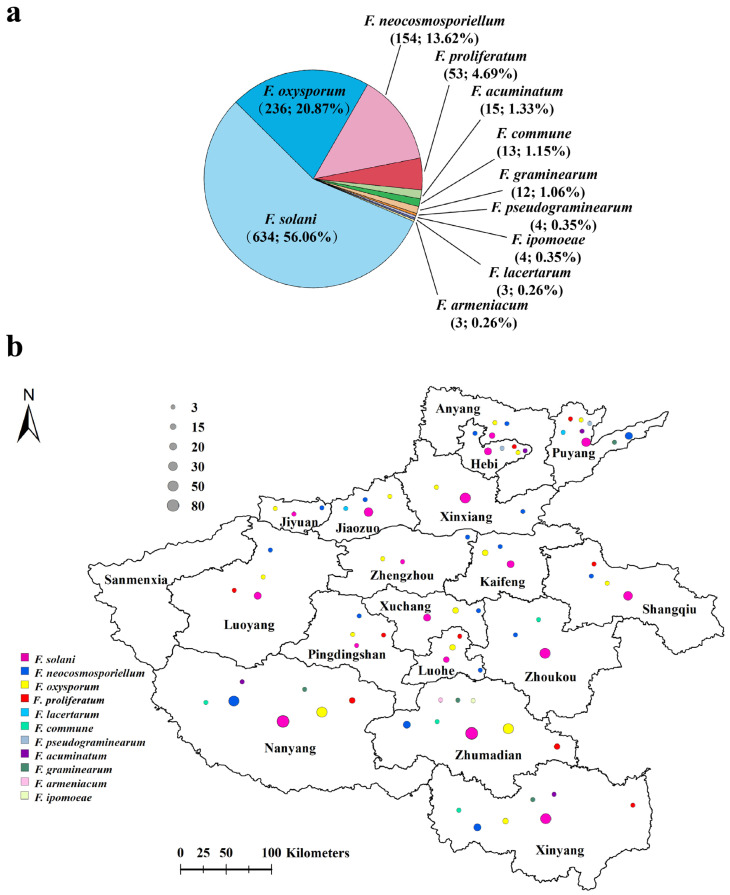
The prevalence of *Fusarium* species isolated from diseased peanut roots. (**a**) The number and overall isolation rate (%) of each *Fusarium* species; (**b**) distribution of fungal species in Henan Province, China, each color represents one fungal species, and the size of the circle represents the number of isolates.

**Figure 6 jof-11-00433-f006:**
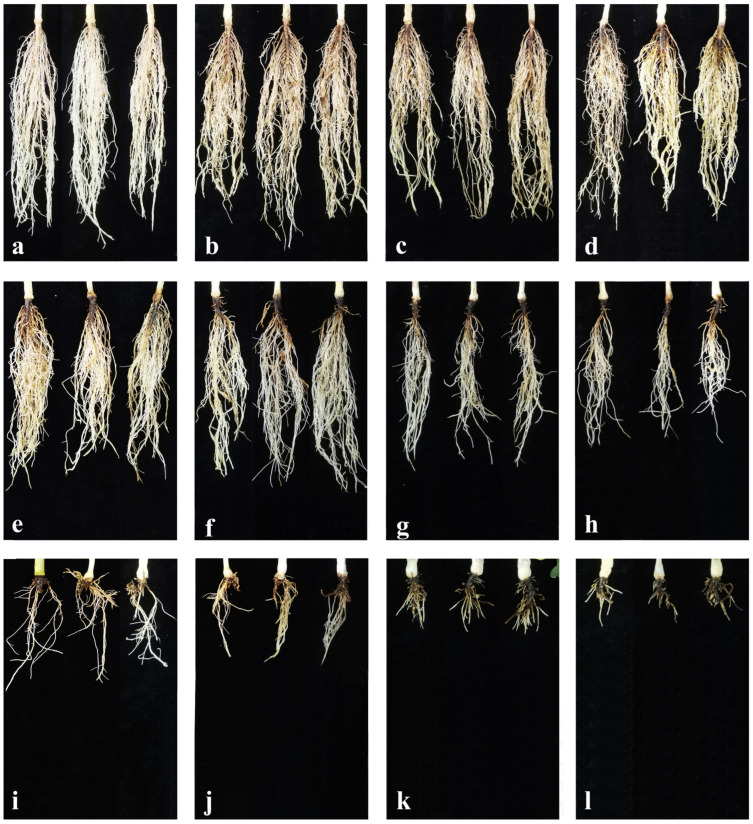
Root rot symptoms on peanuts were observed after 21 days of inoculation with 11 *Fusarium* species through millet seeds. (**a**) Control without inoculation; (**b**) *F. lacertarum*; (**c**) *F. ipomoeae*; (**d**) *F. armeniacum*; (**e**) *F. acuminatum*; (**f**) *F. pseudograminearum*; (**g**) *F. graminearum*; (**h**) *F. commune*; (**i**) *F. solani*; (**j**) *F. oxysporum*; (**k**) *F. neocosmosporiellum*; (**l**) *F. proliferatum*.

**Figure 7 jof-11-00433-f007:**
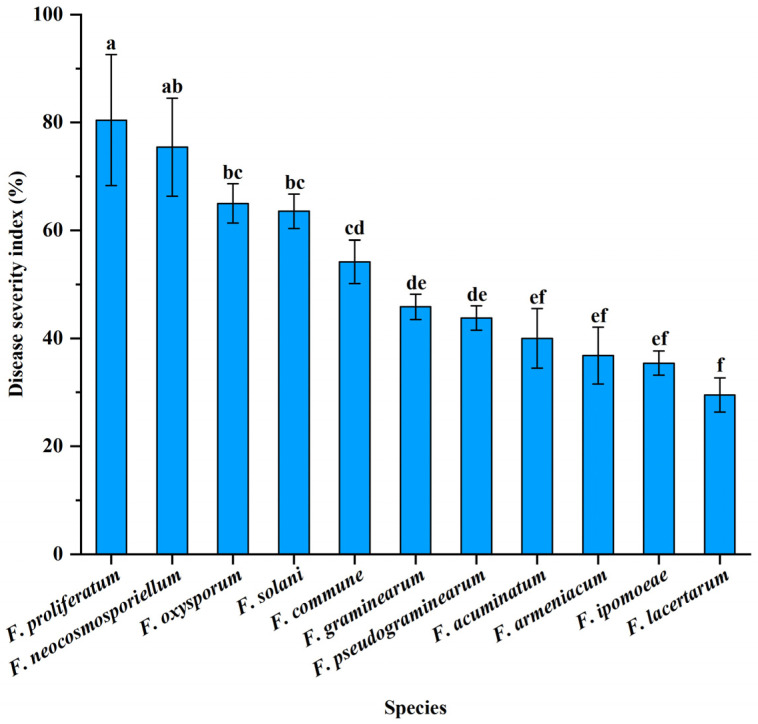
Variation in virulence of the different *Fusarium* species. Means followed by the same letter are not significantly different at *p* = 0.05 based on analysis of variance (ANOVA) and Tukey’s HSD test. Bars and error bars represent means ± SDs.

**Figure 8 jof-11-00433-f008:**
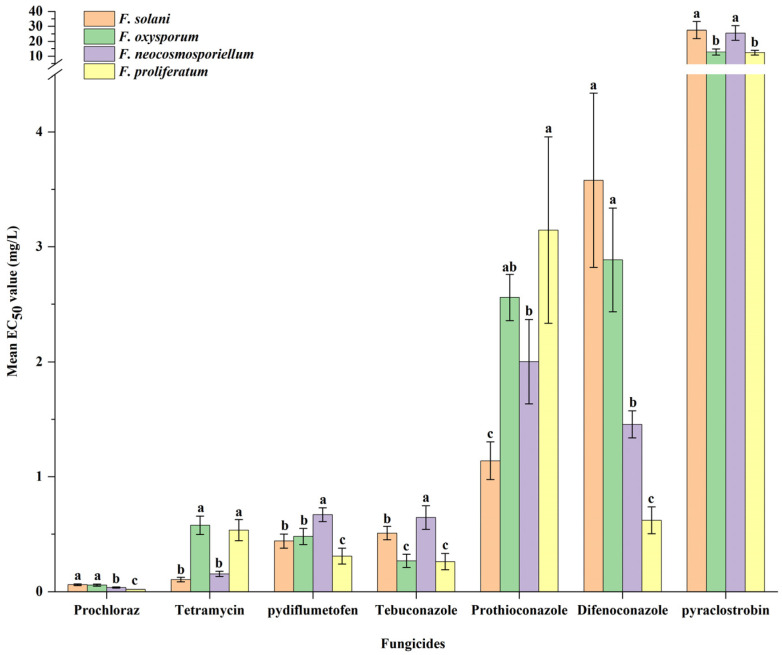
The EC_50_ values of four *Fusarium* species (*F. solani*, *F. oxysporum*, *F. neocosmosporiellum* and *F. proliferatum*) from peanut root to prochloraz, tetramycin pydiflumetofen, tebuconazole, prothioconazole, difenoconazole and pyraclostrobin. *Fusarium* species followed by the same letter are not significantly different at *p* = 0.05 based on analysis of variance (ANOVA) with Tukey’s HSD test. Bars and error bars represent means + SDs (n = 5).

**Table 1 jof-11-00433-t001:** Primers used for PCR and sequencing in the present study.

Locus	Primer	Direction	Sequence (5′ to 3′)
ITS	ITS1	Forward	TCCGTAGGTGAACCTGCGG
	ITS4	Reverse	TCCTCCGCTTATTGATATGC
*TEF-1a*	EF1	Forward	ATGGGTAAGGARGACAAGAC
	EF2	Reverse	GGARGTACCAGTSATCATG
*RPB2*	5f2	Forward	GGGGWGAYCAGAAGAAGGC
	7cr	Reverse	CCCATRGCTTGYTTRCCCAT
*TUB2*	T1	Forward	AACATGCGTGAGATTGTAAGT
	T2	Reverse	TAGTGACCCTTGGCCCAGTTG
*CAM*	CL1	Forward	GARTWCAAGGAGGCCTTCTC
	CL2A	Reverse	TTTTTGCATCATGAGTTGGAC

## Data Availability

The sequences from the present study were submitted to the NCBI database (https://www.ncbi.nlm.nih.gov/, accessed on 25 May 2025). The accession numbers are listed in Appendix A.

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
