# Peer review of "Species Identification and Fungicide Sensitivity of Fusarium spp. Causing Peanut Root Rot in Henan, China"

_jof, 2025, doi:10.3390/jof11060433_

Round 1
Reviewer 1 Report
The work of Min Li and co-authors is devoted to a very urgent problem - the spread of pathogenic Fusarium species that can cause root rot and lead to the loss of the peanut harvest, an important agricultural crop. The study was carried out in the fields of Henan Province (China), where a significant proportion of peanuts are grown. According to the results of the study, 11 species of the genus Fusarium were found, and their susceptibility to various fungicides was also assessed.
I would like to note the high quality and large volume of work performed, the study is large-scale and very useful in terms of controlling fungal diseases.
I recommend the manuscript for publication with minimal edits.
As a reviewer, I have only a few recommendations for improving the article:
1. I would like higher quality figures
2. Perhaps Table 1 should be moved to the Supplementary materials
3. Check the manuscript for typos
Reviewer 2 Report
The manuscript presents important data on Fusarium species associated with PPR, and is overall clearly written and technically sound. However, several key points require clarification and improvement to enhance the quality and scientific value of the study.
Firstly, (1) in Line 57, the authors mention the most common pathogens of PPR, but do not list other fungal species reported in the literature. It would be helpful to clarify whether any of these were also detected in the current study to better understand if PPR is a disease complex involving multiple fungi and the specific role of Fusarium spp.
(2) Morphological identification is based on traits but lacks supporting visual documentation. It is essential to include representative images of macro- and microconidia, conidiophores, and colony morphology for each species to improve transparency and scientific rigor—especially given that over 1,100 isolates were identified morphologically.
(3) The map showing species prevalence should be improved by enlarging it or/and adding pie charts or bar graphs for each region to clearly represent species distribution.
(4) The method used for in vitro sensitivity relies solely on mycelial growth rate, which is not the most representative or reliable model (also please add pictures). Spore germination assays are typically more accurate and should be considered.
(5) Pathogenicity testing on roots should include images of all replicates, rather than selecting a single root as a representative. One of the major challenges in root-based pathogenicity assays is the high degree of variance between replicates. Including all replicates would ensure greater transparency and reliability in visual assessment of pathogenicity, especially when drawing conclusions based on symptom severity or differences between isolates.
6) Regarding sequence data, it is noted that some of the sequence identifiers mentioned (e.g., LM 214915) are not in the correct format or not available in public databases such as NCBI and can not be checked.
These issues are discussed collectively in the major comments section.
Reviewer 3 Report
I have reviewed the submitted manuscript, which investigates the predominant Fusarium species associated with peanut root rot (PRR) in Henan Province, China, and evaluates their sensitivity to various synthetic fungicides. The study provides valuable insights into a significant phytopathological issue, contributing important data on the specific Fusarium species responsible for PRR and the efficacy of commonly used chemical treatments. Given the increasing threat posed by Fusarium spp. to peanut production, these findings are highly relevant to researchers and agricultural practitioners.
While the manuscript presents meaningful results, several areas require improvement to align with the publication standards of the Journal of Fungi. Below, I summarize the key reasons for recommending minor revision:
- Abstract
The abstract is generally clear and follows a logical structure (Background-Objective-Methods-Results-Conclusion). However, some improvements are necessary:
-Lines 13-15: I suggest rewriting it as:
“Peanut Root Rot (PRR) is a devastating disease that significantly limits peanut production worldwide.”
-Lines 17-18: The sentence can be improved as fellow:
“Between 2021 and 2023, we surveyed 81 peanut fields across 17 cities in Henan Province, China, to assess PRR prevalence and Fusarium species distribution.”
-Lines 23-25: The phrasing can be improved for clarity:
“All 11 Fusarium species were capable of causing PRR, with F. solani exhibiting the highest isolation frequency and widespread distribution in all areas.”
However, there is an inconsistency in the claim that F. solani has the highest virulence. According to Figure 6, F. proliferatum exhibits the highest disease severity, indicating greater virulence. This should be corrected to avoid misinterpretation.
Lines 25-28: This paragraph would benefit from additional specificity regarding fungicide effectiveness. Including ECâ‚…â‚€ values would provide quantitative support for the findings.
Additionally, in the Keywords section:
“Predominant species” is too vague and could be replaced with “Fusarium species distribution.”
Instead of “virulence,” the term “pathogenicity” may be more appropriate.
- Introduction
The introduction is well-structured, progressing logically from peanut production to the global impact of PRR, pathogenic diversity, and the study’s justification and novelty. However, several areas could be improved:
-Line 42: The reference (www.stats.gov.cn, accessed on 10 February 2025) should be formatted according to the journal's reference style and moved to the reference section. In-text citations should follow the journal's standard format.
-Line 45: A supporting reference should be added.
-Line 46: The phrase “PRR serves as a devastating disease in peanut” is awkward. It should be reworded as:
“PRR is a devastating disease affecting peanut crops.”
-Lines 48-52: The description of symptoms should be reorganized in a logical progression: Early symptoms, infection progression to advanced infection.
-Lines 57-67: The discussion of Fusarium species distribution should be made more concise by summarizing the key species first, followed by regional variations. A suggested revision:
“Various Fusarium species have been implicated in PRR worldwide, with F. solani and F. oxysporum being the most frequently reported pathogens across multiple regions, including Argentina, …”
-Line 93: The phrasing should be adjusted:
“The main objectives of this research were to (i) identify...” instead of “The main objectives of the research were (i) identify…”
- Material & Methods
This section is well-structured and provides clear methodological details. However, some aspects require clarification and improvement:
-Line 107: The reasoning behind the selection of 15-20 plants per field for sampling is not explicitly stated. It would be useful to justify this choice.
- Line 130: The TEF1-α gene was selected for phylogenetic analysis. Was this the only gene considered sufficient for species identification, or were alternative markers (ITS, RPB2, β-tubulin) evaluated? A brief justification for this choice would be needed.
- Line 164: The reference (http://tree.bio.ed.ac.uk/software/figtree/) should be moved to the reference section and formatted according to the journal’s citation style. In the main text, it should be cited in the standard journal format.
-Line 190: The peanut variety (Yuhua 22) used in the study is specified, which is beneficial. However, additional details on why this variety was chosen would provide more context.
- Results
This section is well-structured and effectively presents different aspects of the study. The figures are satisfactory and appropriately support the findings.
- Discussion
This section is well-structured and effectively presents and discusses the key findings of existing literature. However, some refinements are necessary:
-Line 402: "emerging threat" should be corrected to "an emerging threat."
-Line 403: "needed" should be changed to "need."
-Line 404: "found to have" would be better replaced with "exhibited."
-Line 408: "as present" should be corrected to "at present."
-Line 412: The reference (http://www.icama.org.cn, accessed on 10 February 2025) should be cited correctly in the reference section, with only the appropriate journal format used in the main text.
- Conclusion
This section effectively summarizes the main findings and highlights the novelty of the study, including the prevalence of Fusarium species and their fungicide sensitivity but future perspectives are missing.
- References
The references are up-to-date and include recent research relevant to the paper's field.
Reviewer 4 Report
My concerns were raised above.
L370-371. PLease rephrase 'Koch’s postulates confirmed these 11 Fusarium species as the causative agents of PRR.' Koch's postulates confirm that the isolates are pathogenic.
Reviewer 5 Report
The manuscript presents the results of the study of pathogens causing PRR disease of peanuts in the Hunan province of China. Authors analyzed a large set of samples and identified a number of Fusarium species causing this disease using both morphological evaluation of colonies and DNA sequencing. For each species, the occurrence and virulence were determined. Representatives of all detected Fusarium species were also analyzed for their resistance to 7 different fungicides.
The manuscript is very-well written, has a correct structure and logic. Intro section describes the situation with the PRR disease in China and the current level of studies in this field and substantiates the purpose of the study. Materials and Methods section includes a thorough description of approaches used in the study. The Results section comprehensively describe the obtained results presented in diagrams and tables. Discussion section reviews the obtained results and compares them with the results of other researcher as well as explains some discrepancies and formulates the further prospects for use of the obtained results. Conclusions are supported by the results.
I would want to thank authors for the competent planning and realization of their experiments, thorough preparation of this manuscript, and clear presentation and discussion of the obtained results. The study design is excellent. Also, the language of the manuscript is excellent and almost do not require any editing.
I have only two very minor comments:
Line 108: please, give a reference for a 5-point sampling approach or describe it.
Line 337-338: ...fungicides...were divided into five classes (...) were used... Please, check the sentence and correct.
Round 2
Reviewer 4 Report
All major concerns were addressed by authors.
All major concerns were addressed by authors.